# Exploring commercial sex work among transgender women in Nepal: Contributors and stigma—A mixed-method study

Himani Shree Pokharel[1]◉, Salau Din Myia[1], Anisha Chalise[2], Shishir Paudel◉[1]◉*

1 Department of Public Health, CiST College, Pokhara University, Kathmandu, Nepal, 2 Center for Research on Environment, Health and Population Activities (CREHPA), Lalitpur, Nepal

◉ These authors contributed equally to this work.
* shishirpaudel11@gmail.com

**Data Availability Statement:** All relevant data are within the manuscript.

## Abstract

### Introduction

Transgender women are considered as one of the most vulnerable populations for commercial sex work and sexually transmitted infections. This mixed-method study aimed to explore the factors associated with commercial sex work among transgender women of Nepal along with their experience and the stigma associated with it.

### Methods

A concurrent mixed-methods approach was employed in this cross-sectional study conducted from July to December 2022 in the Kathmandu Valley of Nepal. A total of 152 transgender women participated in the quantitative survey, and 17 of them were selected for in-depth interviews (IDIs). The chi-square test was employed at a 5% level of significance to identify factors associated with commercial sex work. Qualitative data from the IDIs were thematically analyzed, with key themes emerging around reasons for engaging in sex work and the stigma associated with it including the Silence, Shame, and Dignity & Treatment domains under sex work stigma.

### Results

Out of 152 transgender women, 104 (64%; 95% CI: 60.5–75.0%) were found to be currently working as a commercial sex worker. Factors such as education, household wealth, homelessness, history of sexual and verbal abuse, including rape and physical attacks, and socialization with others were found to have a statistically significant relation with commercial sex work at p<0.05. Despite engagement in commercial sex work, a higher proportion experienced sex work stigma under the Sex Work Stigma Scale.

### Conclusion

The findings, derived from both quantitative and qualitative studies, emphasize the urgency of targeted interventions to alleviate the challenges faced by this marginalized population.

**Funding:** The author(s) received no specific funding for this work.

**Competing interests:** The authors have declared that no competing interests exist.

The observed higher proportion of commercial sex work and sex work stigma underscores the pervasive social challenges of transgender women. A comprehensive and coordinated effort is essential, bringing together governmental, private sector, and public awareness initiatives to effectively manage stigma and address the underlying causes of commercial sex work.

## Introduction

Transgender women are those individuals assigned male-at-birth but self-identify themselves as female [1]. Globally, the transgender population is expected to account for 0.1% to 1.1% of the total population of reproductive age, yet this figure could be imprecise due to the uneven "case" definition of transgender identity and pervasive stigma associated with it [2,3]. As a consequence of this social stigma and unacceptability, transgender individuals experience discrimination, social exclusion, and violation of their general human rights leading to social, economic, and health vulnerabilities [2,4]. In the context of Nepal, the government officially recognized the third gender category in 2007 and has introduced protective laws for gender and sexual minorities recognizing their equal rights [5]. The Constitution of Nepal 2015, under articles 12, 18, and 42 aims to guarantee fundamental rights for Lesbian Gay Bisexual Transgender Queer Intersex Asexual (LGBTQIA+) individuals [6]. However, a noteworthy gap persists between legal recognition and the ground realities. The recent census of 2021, reveals a total of 2,928 individuals identifying themselves as sexual and gender minorities, i.e., other gender, accounting for 0.01% of the total population, with disproportionate distribution across the seven provinces of Nepal [7].

Sex work, which is the exchange of sexual services or favors for monetary payment [8]. A systematic review of 29 studies based in the US, indicates significant vulnerability of transgender women to engage in commercial sex work [9], with a lifetime engagement rate ranging from 24% to 75% across countries of different economies [10]. This vulnerability stems from the intersection of unemployment, underemployment, and workplace mistreatment, because of their gender identity [11], which could potentially lead them towards commercial sex work for income generation and as a means to fulfill their basic needs such as food, rent, and shelter [4,12,13]. Transgender sex workers, in general experience widespread violence and rights violations due to the intersection of transgender stigma, sex work stigma, and other marginalized identities [14–17]. A systematic review and meta-analysis based on ten low and middle-income countries noted a disproportionate impact of HIV among transgender women as 17.5% were living with HIV [18]. In comparison to cis-gendered women sex workers, transwomen sex workers were found to have nine-fold increase in the risk of contracting HIV infection [19], illustrating significantly high vulnerability to sexually transmitted infections among transwomen sex workers. Although commercial sex work is considered an act of trafficking in Nepal which is punishable under the Human Trafficking and Transportation (Control) Act 2008 and National Penal Code 2074, its prevalence still persists [20,21].

In recent years, multiple national reports and news articles from Nepal have highlighted the involvement of transgender individuals in commercial sex work, but a critical research gap remains in exploring the magnitude of this problem, its contributing factors, and the stigma related to it in this vulnerable group [21–23]. In this regard, this mixed-method study aimed to explore the factors associated with commercial sex work among transgender women of Nepal along with their experience and stigma associated with it.

## Materials and methods

### Study design and setting

A concurrent mixed method (quantitative and qualitative) cross-sectional study design was employed among randomly selected transgender women residing in Kathmandu Valley from July to December 2022. Kathmandu Valley of Nepal lies in Bagmati Province, covering three urban districts of Nepal namely Kathmandu, Bhaktapur, and Lalitpur Districts, with a total population of 3,025,386 accounting for 10.37% of the National Population [7]. It is estimated that almost 958 individuals identifying as other gender reside in Bagmati province of Nepal, which represents 32.7% of the total population of these marginalized population sub-groups [7]. However, Blue Diamond Society, which is a non-governmental organization working for the empowerment of sexual and gender minorities in Nepal, proclaims this number to be at least a thousand inside Kathmandu Valley alone [24].

### Sampling procedures and participants' recruitment

The sample size for the quantitative survey was determined using Cochran's formula for the estimation of a proportion (n = $z^2pq/d^2$) adjusted for a finite population. A past study based in Kathmandu Valley reported the lifetime prevalence of commercial sex work among transgender individuals to be 73.4% [25]. Considering this past prevalence at 5% allowable error and 95% confidence interval, the sample size was estimated to be 152. Due to the hidden nature of the study population, the snowball sampling technique was adopted, where transgender women were requested to provide the contact details of other transwomen and the process continued until the required sample size was met. The recruitment process began with five initial seeds, who were transgender women engaged in social or activist networks. These seeds were asked to refer other transgender women they knew who fit the study's eligibility criteria. Through this process, four referral chains were formed to capture the full sample. All the transgender women aged 18 and above, residing inside Kathmandu Valley during the time of data collection were eligible to participate in this study. The transgender women who were not residing in Kathmandu Valley during the time of data collection, those below 18 years of age, and those who were unwilling to participate were excluded.

From among the participants of the quantitative survey, a total of 17 transgender women were purposively selected and approached for the in-depth interviews (IDI) based on their willingness to be interviewed as well as their socioeconomic, and professional roles. The number of IDIs was determined using the principle of data saturation, where interviews continued until no new information or themes emerged from subsequent interviews. Data saturation was reached after 17 interviews, as no new themes were being identified in the last few interviews.

### Data collection

The data were collected through face-to-face interviews for both the quantitative and qualitative phases of this study. As the participants were a hard-to-reach population, both quantitative and qualitative interviews were carried out concurrently to optimize recruitment and participation. During the quantitative interviews, participants found to be engaged in commercial sex work were asked if they would be willing to participate in the qualitative phase.

For the quantitative survey, a structured questionnaire consisting of close-ended questions was used. The questionnaire was divided into three sections. The first section included questions regarding participants' socio-demographic characteristics including age, education, household wealth, and relationship status. The second section consisted of questions regarding their interpersonal factors such as past experiences of sexual & verbal abuse and socialization

with other trans individuals. The third section consisted of the Experiences with Sex Work Stigma Scale (ESWS) [26], a standard tool to assess sex work-related stigma among commercial sex workers. The ESWS scale consists of 27 items covering four domains, including silence domain, shame domain, dignity domain and treatment domain. The ESWS scale was administered to those participants who reported being engaged in commercial sex work. ESWS was translated into the Nepali language, maintaining the translation validity through translation and back translation (English-Nepali-English). The complete quantitative survey tool was pretested among 30 transgender women of Bharatpur Municipality, Chitwan, currently engaged in commercial sex work to ensure the reliability and validity of the tool. The inter-item reliability of ESWS was noted to be satisfactory with Cronbach's alpha of 0.88. On average, the quantitative survey interview took 30 minutes.

For the qualitative phase, an open-ended interview schedule was employed to gather in-depth insights into participants' lived experiences and stigmas related to commercial sex work. The IDIs typically lasted around 30 to 45 minutes. The interview guideline for IDI was developed by HSP under the guidance of SP and AC. The guideline was prepared based on the observations made during pretesting of quantitative tools. The expert opinion on the guideline was provided by SDM and the representatives of the trans community and the Blue Diamond Society. It was piloted among two transgender women and was revised based on their feedback.

All interviews for both the quantitative and qualitative phases were conducted in person by HSP at the residency of the transgender women, following the acquisition of written informed consent from the participants, along with their permission to audio record the IDIs. To facilitate open discussion of sensitive topics, all interviews were held in private settings. All the interviews were conducted by HSP, a public health undergraduate, who also took field notes during each session. HSP was trained by senior researchers (SP and AC) in ethical considerations, rapport building, maintaining participant confidentiality, managing emotional distress during interviews, and employing techniques for conducting qualitative in-depth interviews.

## Data management and analysis

Quantitative data, entered using EpiData version 3.1 was exported to Statistical Package for Social Sciences version 22 for analysis. Descriptive statistics were used to describe participants' demographic profiles, interpersonal factors, and domains related to sex-work stigma. Chi-square test was performed at a 5% level of significance to identify the factors associated with commercial sex work among transgender women.

The audio recordings of IDIs were transcribed in Nepali and then translated to English by three authors (HSP, AC, SP). The transcripts were cross-checked for accuracy and language translation consistency from Nepali to English. Prolonged engagement of authors in every phase of the study and peer debriefing for translations of verbatim were done. Due to the sensitive nature of the study population and their time constraint, the transcripts were not sent to the participants for their review. The Braun and Clark's six-step thematic analysis with inductive-deductive continuum was used for qualitative data analysis [27]. The interview transcripts and notes were read and analyzed several times to understand the emotions and experiences of participants and to become thoroughly familiar with the content. The qualitative data was analyzed manually, without using any data coding and analysis software. Based on the information provided by the participants, the core concepts expressed by each of the participants were extracted as codes and further clustered into themes. All authors collaboratively reviewed and finalized the themes. The initial codes were prepared by the first author, who is a public health undergraduate. The codes were reviewed and clustered into themes by AC and SP, who are

public health postgraduates with past experiences in both quantitative and qualitative data collection and analysis. The overall process was supervised by SDM, a public health expert with years of experience in public health research.

### Ethical consideration

The ethical approval for this study was obtained from the Institutional Review Committee of CiST College (Registration no: IRC/154/078/079). Written informed consent was obtained from all the participants prior to data collection in both the quantitative and qualitative phases. The participants of the IDIs were also asked for their permission to audio-record the sessions. Participant privacy and confidentiality was maintained throughout, and all identifying information was anonymized using participant codes.

## Results

Among the total of 152 transgender women approached for this study, the overall prevalence of commercial sex work was observed to be 68.4% (95% CI: 60.5–75.0%) as 104 reported working as commercial sex workers. The age of the participants ranged from 18 to 54 years, where the mean age was 27.84±0.71 years. In context of the education level of the participants, only 13.2% had a bachelor's degree or higher education. In regards to socialization, about half of the participants (49.3%) reported socializing through political activism whereas more than two-fifths (61.2%) socialized through support groups. Alarmingly, more than a fifth (22.4%) of the participants reported to have experienced homelessness in their lifetime. More than three-quarters (77.6%) of the participants had experienced sexual abuse, and around three-fourths of them (67.8%) were abused before the age of 14. The chi-square test revealed that socio-demographic and interpersonal factors such as education, household wealth, homelessness, history of sexual and verbal abuse including rape and physical attacks as well as socialization status were associated with commercial sex work among transgender women (Table 1).

### Stigma associated with sex work

The ESWS scale was used to assess the stigma associated with sex work. Out of 104 commercial sex workers, about four-fifths (82.7%) reported keeping their occupation a secret. Nine out of ten (92.3%) reported feelings of rejection and exclusion due to their work. Similarly, nine out of ten (90.3%) reported never being accepted due to their occupation. More than half (55.7%) experienced humiliation by others (Table 2).

The characteristics of the 17 transgender women who participated in the in-depth interviews is illustrated in Table 3. These participants were purposively selected based on their socio-economic and professional diversity to capture a range of experiences.

### Reasons for engagement in commercial sex work

The participants in IDIs were asked about the reasons for engaging in commercial sex work, where most transgender women suggested that they got involved in it mostly due to the lack of other economic options as they are forced to drop out of educational institutions, denied job opportunities, and had to deal with homelessness which makes them economically vulnerable and compels them to sell their bodies for survival. However, the participants also expressed that working as a sex worker has provided them with an opportunity to express their individuality while providing them financial freedom. Some participants also reported quitting commercial sex work after gaining financial independence along with elevated social status.

**Table 1. Factors associated with commercial sex work (n = 152).**

| Variable | n (%) | Commercial sex work | | Chi-square | p-value |
|---|---|---|---|---|---|
| | | Yes (%) | No (%) | | |
| **Age** | | | | | |
| <25 years | 68 (44.7) | 52 (76.5) | 16 (23.5) | 4.437 | 0.109 |
| 25–35 years | 62 (40.8) | 40 (64.5) | 22 (35.5) | | |
| ≥35 years | 22 (14.5) | 12 (54.5) | 10 (45.5) | | |
| **Level of education** | | | | | |
| Primary level | 26 (17.1) | 24 (92.3) | 2 (7.7) | 24.585 | <0.001** |
| Secondary level | 106 (69.7) | 75 (70.8) | 31 (29.2) | | |
| Bachelor and above | 20 (13.2) | 5 (25) | 15 (75) | | |
| **Household wealth** | | | | | |
| Extremely poor | 29 (19.1) | 22 (75.9) | 7 (24.1) | 23.095 | <0.001** |
| Poor | 37 (24.3) | 34 (91.9) | 3 (8.1) | | |
| Middle class | 24 (15.8) | 16 (66.7) | 8 (33.3) | | |
| Upper middle class | 31 (20.4) | 20 (64.5) | 11 (35.5) | | |
| Rich | 31 (20.4) | 12 (38.7) | 19 (61.3) | | |
| **History of sexual abuse** | | | | | |
| Yes | 118 (77.6) | 87 (73.7) | 31 (26.3) | 6.878 | 0.009* |
| No | 34 (22.4) | 17 (50) | 17 (50) | | |
| **Age at abuse** | | | | | |
| ≤14 years | 80 (67.8) | 54 (67.5) | 26 (32.5) | 4.976 | 0.026* |
| >14 years | 38 (32.2) | 33 (86.8) | 5 (13.2) | | |
| **History of rape** | | | | | |
| Yes | 80 (52.6) | 66 (82.5) | 14 (17.5) | 15.494 | <0.001** |
| No | 72 (47.36) | 38 (52.8) | 34 (47.2) | | |
| **Experience of verbal abuse** | | | | | |
| Yes | 131 (86.2) | 97 (74) | 34(26) | 13.884 | <0.001** |
| No | 21 (13.8) | 7 (33.3) | 14 (66.7) | | |
| **Experience of physical attack** | | | | | |
| Yes | 109 (71.7) | 92 (84.4) | 17 (15.6) | 10.496 | 0.001* |
| No | 43 (28.3) | 12 (27.9) | 31 (72.1) | | |
| **Relationship status** | | | | | |
| Single | 96 (63.2) | 63 (65.6) | 33 (34.4) | 0.956 | 0.620 |
| In a relationship and living together | 41 (27.0) | 30 (73.2) | 11 (26.8) | | |
| In a relationship but not living together | 15 (9.9) | 11 (73.3) | 4 (26.7) | | |
| **Socialization with other trans-individuals by political-activism** | | | | | |
| Yes | 75 (49.3) | 64 (85.3) | 11 (14.7) | 19.599 | <0.001** |
| No | 77 (50.7) | 40 (51.9) | 37 (48.1) | | |
| **Socialization with other trans individuals through support groups** | | | | | |
| Yes | 93 (61.2) | 75 (80.6) | 18 (19.4) | 16.570 | <0.001** |
| No | 59 (38.8) | 29 (49.2) | 30 (50.8) | | |
| **Homelessness** | | | | | |
| Yes | 34 (22.4) | 31 (91.2) | 3 (8.8) | 10.496 | 0.001* |
| No | 118 (77.6) | 73 (61.9) | 45 (38.1) | | |

*Statistical significance at p<0.05

** Statistical significance at p<0.001.

**Table 2. Stigma associated with commercial sex work among transgender women (n = 104).**

| Silence Domain | Always n(%) | Sometimes n(%) | Never n(%) |
|---|---|---|---|
| Tried to make sure no one knows you do sex work | 83 (79.8) | 12 (11.5) | 9 (8.7) |
| You have done everything you can to keep sex work a secret | 86 (82.7) | 9 (5.9) | 9 (5.9) |
| Avoid talking about sex work | 82 (78.8) | 12 (11.5) | 10 (9.6) |
| Conceal working as a sex worker with family | 79 (76.0) | 15 (14.4) | 10 (9.6) |
| Conceal working as a sex worker with community | 86 (82.7) | 12 (11.5) | 6 (5.8) |
| Deny working as a sex worker | 84 (80.8) | 12 (11.5) | 8 (7.7) |
| **Shame Domain** | | | |
| Felt ashamed | 79 (76.0) | 13 (12.5) | 12 (11.5) |
| Felt rejected | 96 (92.3) | 3 (2.9) | 5 (4.8) |
| Felt excluded | 96 (92.3) | 6 (5.8) | 2 (1.9) |
| Felt different | 103 (99.0) | 0 (0.0) | 1 (1.0) |
| Felt humiliated | 86 (82.7) | 17 (16.3) | 1 (1.0) |
| Felt frustrated | 43 (41.3) | 56 (53.8) | 5 (4.9) |
| **Dignity Domain** | | | |
| Felt valued | 2 (1.0) | 20 (19.4) | 82 (79.6) |
| Felt comfortable | 12 (11.5) | 74 (71.2) | 18 (17.3) |
| Felt proud | 7 (6.8) | 7 (6.8) | 89 (86.4) |
| Felt accepted | 5 (4.9) | 5 (4.9) | 93 (90.3) |
| Felt at peace | 7 (6.7) | 18 (17.3) | 79 (76.0) |
| Felt happy | 8 (7.7) | 68 (65.4) | 28 (26.9) |
| **Treatment Domain** | | | |
| Distanced themselves from you | 52 (50.0) | 39 (37.5) | 13 (12.5) |
| Criticized you | 65 (62.5) | 32 (30.8) | 7 (6.7) |
| Excluded you from groups | 36 (34.6) | 57 (54.8) | 11 (10.6) |
| Humiliated you | 58 (55.8) | 42 (27.6) | 4 (3.8) |
| Laughed at you | 90 (86.5) | 14 (13.5) | 0 (0.0) |
| Called you names | 87 (83.7) | 17 (16.3) | 0 (0.0) |
| Ignored you | 41 (39.4) | 63 (60.6) | 0 (0.0) |
| Mistreated you | 34 (32.7) | 65 (62.5) | 5 (4.8) |
| Treated you differently from other women | 97 (93.3) | 7 (6.7) | 0 (0.0) |

"*The lack of support from my family forced me to leave home and became homeless at the age of 10 and I started working as a sex worker by the time I was 13.*"–P1

"*Nobody wants to live like this and sell their body to survive. But what other options do we have? We aren't welcome in offices, schools, and places of business, just to name a few.*"–P2

"*I saw my friends living the way they wanted while working as a sex worker and I was always too afraid to come out, but seeing how free my friends felt, I decided to give it a try.*"–P3

"*I used to be a sex worker but after I started my own business and started getting socially active, I left that profession.*"–P14

## Silence domain

The participants reported hiding their work from their loved ones as they feared being disowned by their families and society. They also expressed being silenced even when they were physically and sexually harassed, as sometimes the abusers were people in power, and at other

Table 3. Characteristics of participants involved in the IDIs (n = 17).

| Participant No. | Age | Education | Commercial sex work status |
|---|---|---|---|
| 1 | 29 | Primary Level (1–5 Grade) | Current |
| 2 | 22 | Higher Secondary Level (11–12 Grade) | Current |
| 3 | 29 | Higher Secondary Level (11–12 Grade) | Current |
| 4 | 28 | Higher Secondary Level (11–12 Grade) | Current |
| 5 | 31 | Primary Level (1–5 Grade) | Current |
| 6 | 21 | Secondary Level (6–10) | Current |
| 7 | 35 | Higher Secondary Level (11–12 Grade) | Current |
| 8 | 24 | Higher Secondary Level (11–12 Grade) | Current |
| 9 | 22 | Primary Level (1–5 Grade) | Current |
| 10 | 21 | Higher Secondary Level (11–12 Grade) | Current |
| 11 | 25 | Undergraduate | Current |
| 12 | 23 | Higher Secondary Level (11–12 Grade) | Current |
| 13 | 29 | Primary Level (1–5 Grade) | Former |
| 14 | 24 | Higher Secondary Level (11–12 Grade) | Former |
| 15 | 19 | Secondary Level (6–10) | Current |
| 16 | 26 | Secondary Level (6–10) | Former |
| 17 | 27 | Higher Secondary Level (11–12 Grade) | Former |

times, they feared revealing their transgender identity as well as their occupation. On the contrary, some individuals disclosed being happy, and much more satisfied with their lives after engaging as commercial sex workers as they did not have to suppress their identity to fit in (Fig 1).

*"If my family knew about my work, they would kill me. They don't even know that I am transgender."* -P4

*"I have been sexually harassed multiple times. I was brutally gang raped by three individuals in positions of authority and almost beaten to death. They knew I was trans, so they raped me because no one would believe me, and I would not even file a complaint."*–P5

*"When I was eighteen, I met a guy through Facebook. He knew my identity but he still wanted to meet me. I was overwhelmed at first, but then he abused me physically and sexually. I was abused and mistreated but was unable to tell even my family."*–P6

*"I lived in fear for too long. I am tired of hiding. I was a school teacher for nearly a decade, but I was never satisfied with my life; I always suppressed my identity to fit in, but now that I am loud and proud, I am much more happy and content."*–P7

## Shame domain

Majority of the IDI participants reported being ashamed of working as a commercial sex worker and thus concealed the nature of their work, all while distancing themselves from their friends and family. They even refrained from talking about sex work and despised being a sex worker (Fig 1).

*"Nobody envisions themselves having a career in sex work, and neither did I, but here I am! My family thinks that I work in sales and I want to keep it that way."*–P8

**Fig 1. Sex work stigma experienced by the participants.**

*"There is not a single day that I don't cry myself to sleep. I hate living like this. I hate hiding my identity and profession from my friends, family, and, most importantly, myself."*–P9

*"Sex work is a burden; nobody wishes for this. If there is a next life, I hope everything will be normal, and I won't be trans. I don't want my family to be ashamed of who I am."*–P10

### Dignity & treatment domain

The dignity and treatment domains were observed to intertwine to a large extent as the treatment the participants received from others around them impacted the degree to which they felt happy, proud, comfortable accepted, or valued. Those transgender women who did not hide their identity were subjected to prejudice, discrimination, and inhumane treatment; being laughed at, called names, and also vulnerable to physical, emotional, and sexual harassment. They were denied job opportunities even though they were capable; were harassed, mistreated, treated as objects, and excluded, not only by strangers but also by their friends and family alike. Nevertheless, some shared that living life on their own terms allowed them to live with dignity and pride. An intriguing case was observed during the IDI, where one participant revealed that the mistreatment that she used to experience for being a trans woman gradually diminished when she assumed the role of a '*Mata*' (one possessed by the divine), but it was only limited to her disciples (Fig 1).

*"I was promised work as a dishwasher, but when I went there, the guy said, "Sleep with me, and then I'll give you the job". I was treated as an object and not a human being because I was a transgender sex worker. And this wasn't even the first time something like this had happened."*–P2

*"Our dignity is non-existent when we are openly trans and sex workers. I've been laughed at, called names, beaten, thrown stones at, stabbed, chased and raped. I have gotten used to it by now."*–P11

*"I was rejected for regular jobs even though I met all the requirements because I was openly trans and a sex worker."*–P12

*"I was a sex worker but then one day Mother Daxinkali (a goddess) appeared in my dreams and then I turned into a 'Mata'. After that day, I started noticing changes in myself. I started to show psychic abilities, and the people who trusted my prophecies started to respect and accept me."*–P13

*"I lived a lie for way too long. So, even if I'm a sex worker, I am happy with where I am in life. I am able to express myself, live with my head held high, and be the woman I was always meant to be."* –P7

### Discussions

In this study, nearly seven out of ten (68.42%) transgender women were found to be engaged in commercial sex work. A similar finding was shared by a past study from Kathmandu Valley in 2019 where about three-fifths (58.4%) of the trans individuals were found to be involved in commercial sex work [25]. Likewise, in neighboring country India, eight out of ten transgender individuals were found to be involved in sex work [28]. A similar higher proportion of

trans individuals' involvement in commercial sex work has been observed by studies throughout the world, regardless of the national economy [10,29,30]. These findings illustrate the vulnerability of trans individuals to various health issues as a result of engagement in commercial sex work, which is becoming a global public health problem.

It was observed that education has a statistically significant relationship with transwomen's engagement in commercial sex work. Almost all (92.3%) of trans women who failed to complete their primary education were involved in commercial sex work. During IDI, it was revealed that shame, stigma, and mistreatment associated with transgender identity led many to drop out of school, resulting in poor education which further complicated their economic opportunities. Similar phenomena were observed in past studies from Nepal, India, Jamaica, and the US, where individuals settled on becoming sex workers as a result of poor education, which further narrowed their employment opportunities [11,31–33]. The harassment, bullying, and social exclusion associated with transgender identity might force these individuals to discontinue their schooling resulting in poor qualifications and competencies forcing them to resort to sex work for survival.

It was observed that almost two-fifths of trans women fall under the poor or extremely poor quintile. The social stigma associated with trans identity was found to have prevented these individuals from obtaining and retaining jobs, making it hard for them to seek better economic opportunities. It was also observed that commercial sex work somehow provided these vulnerable individuals a sense of economic independence, attracting more trans individuals to commercial sex work. These observations have been shared by past studies from different developing as well as developed nations. A cross-sectional study from the US noted that the rates of poverty among trans individuals are higher as their household incomes are exceptionally lower [34]. Similarly, studies from India revealed that trans individuals from Kolkata and Odisha were being deprived of formal employment due to their trans-identity, and poor qualifications, and even those engaged in a formal sector, were forced to quit due to workplace harassment [35]. Furthermore, in this study, more than one-fifth of the participants reported experiencing homelessness at some point in life, mostly due to their gender identity crisis, which forced them to engage in commercial sex work. In some cases, it was also observed that with the improvement in the social and financial status of this vulnerable population, they tend to distance themselves from this profession. Thus, the empowerment of trans women and ensuring their financial independence plays a critical role in preventing them from being forced into commercial sex work.

It was observed that being verbally abused and physically attacked are some of the factors significantly associated with commercial sex work. This finding is corroborated by various other studies suggesting that transgender individuals experience elevated levels of verbal abuse, threats of violence, and intimidation along with physical violence, the majority of which is a result of their gender identity and commercial sex work [11,36]. Over three-fourths of the participants involved in sex work alarmingly had a history of sexual abuse, and nearly three-quarters were victimized before the age of fourteen, even before engaging in commercial sex work. The IDIs reflected that transgender women frequently get sexually abused by family members, relatives, and police authorities which leads them towards isolation and homelessness. This somehow contributes to transgender women resorting to commercial sex work, where they further get abused by strangers and their clients. This finding is in line with various studies in the countries of the Asia Pacific region and also the US, where transgender women were found to be commonly violated by their clients and police officers and these cases were often unreported [11,37,38]. A multi-country study based in Asia also reported the elevated rates of rape and victimization of transgender individuals engaged in sex work [36]. During IDI, the participants disclosed being raped by their clients, relatives, someone they know, and

even the police as a common repercussion of being a sex worker and seemed to have accepted it as their fate, as little to no actions were taken against the perpetrators. Furthermore, a notable proportion of stigmatization was seen across different domains of the sex work stigma scale. The IDIs further highlighted that stigmatization has negatively impacted the mental and emotional well-being of transgender women engaged in sex work. This illustrates the need for interventions to prevent stigma and discrimination associated with trans-gender identity and sex work and also to protect their emotional and mental well-being.

The findings of this study highlight the need for comprehensive policy interventions to improve the lives of transgender women in Nepal including those involved in commercial sex work. Education, homelessness and poor economic status were found to be associated with involvement in commercial sex in quantitative data which was further support by qualitative findings as school dropout and lack of other economic opportunities and maltreatment at workplace were root causes driving transgender women into commercial sex work. Thus, inclusive education policies are essential to prevent school dropout among LGBTQIA+ individuals by addressing gender-based stigma and bullying. Providing scholarships and enforcing anti-bullying measures would empower transgender individuals, expanding their economic opportunities beyond commercial sex work. Nepal's Constitution of 2015 is inclusive of LGBTQI identities where Article 18 (3), guarantees the right to equality and non-discrimination to "gender and sexual minorities" [6]. However, stronger enforcement and expansion of these protections are needed to ensure transgender individuals are not excluded from formal employment due to their gender identity. Additionally, targeted vocational training and employment initiatives should be created to help transgender women currently engaged in commercial sex work achieve economic independence and transition to alternative livelihoods. Furthermore, policies addressing housing security and mental health support are critical, as homelessness and mental distress were key issues faced by transgender women, often pushing them into commercial sex work.

This is one of the few studies that have assessed the prevalence of commercial sex work and its associated factors among transgender women of Nepal, along with their lived experiences and stigma related to sex work. However, the study is not free from its limitations, and the interpretations must be made acknowledging these limitations. Despite the efforts to cover diverse participants for better representation, due to the hidden nature of the study population, non-probability sampling was adopted which might have introduced some selection bias. Regardless of our efforts to protect the privacy and confidentiality of the participants and their data, the sensitive nature of research questions could have introduced some social desirability bias. Although this study was executed in the national capital for better representation as a higher proportion of transgender individuals reside here, yet, Nepal being culturally and socially diverse, further studies focusing on these vulnerable populations in rural areas of Nepal might be fruitful. Future research could benefit from employing a larger and more diverse sample to allow for a clearer exploration of how various intersecting factors, such as socioeconomic status, ethnicity and age shape the experiences and outcomes of the study population.

## Conclusions

The research sought to investigate factors associated with commercial sex work among transgender women living in Kathmandu Valley, Nepal, along with the associated stigma. The findings, derived from both quantitative and qualitative studies, revealed a significant association of commercial sex work with factors such as education, economic status, experiences of sexual abuse and harassment, socialization with other transgender individuals, and homelessness. It

is crucial to address the repercussions of these factors to promote the mental health and overall well-being of transgender women. To effectively manage the stigma and underlying causes of commercial sex work, a coordinated effort involving government, private sector, and public awareness at all levels is imperative.

## Supporting information

**S1 Checklist. Supplementary File 1: COREQ checklist.**
(DOCX)

**S2 Checklist. STROBE statement—checklist of items that should be included in reports of *cross-sectional studies*.**
(DOCX)

## Acknowledgments

We share our gratitude to all the transgender women who participated in this study and provided their valuable time and information. Without them, this study wouldn't have been possible. We are thankful to the Blue Diamond Society and its representatives for their constant support during the study.

## Author Contributions

**Conceptualization:** Himani Shree Pokharel, Salau Din Myia, Anisha Chalise, Shishir Paudel.

**Data curation:** Himani Shree Pokharel, Anisha Chalise, Shishir Paudel.

**Formal analysis:** Himani Shree Pokharel, Anisha Chalise, Shishir Paudel.

**Investigation:** Himani Shree Pokharel.

**Methodology:** Himani Shree Pokharel, Anisha Chalise, Shishir Paudel.

**Project administration:** Himani Shree Pokharel, Salau Din Myia, Shishir Paudel.

**Resources:** Himani Shree Pokharel.

**Supervision:** Salau Din Myia, Shishir Paudel.

**Validation:** Himani Shree Pokharel, Salau Din Myia, Anisha Chalise, Shishir Paudel.

**Visualization:** Himani Shree Pokharel, Anisha Chalise, Shishir Paudel.

**Writing – original draft:** Himani Shree Pokharel, Anisha Chalise, Shishir Paudel.

**Writing – review & editing:** Himani Shree Pokharel, Anisha Chalise, Shishir Paudel.

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
