## [Decision Letter · Decision Letter 0]

24 Sep 2024

PONE-D-24-19414Exploring Commercial Sex Work among Transgender Women in Nepal: Contributors and Stigma - A Mixed-Method StudyPLOS ONE

Dear Dr. Paudel,

Thank you for submitting your manuscript to PLOS ONE. After careful consideration, we feel that it has merit but does not fully meet PLOS ONE’s publication criteria as it currently stands. Therefore, we invite you to submit a revised version of the manuscript that addresses the points raised during the review process.

Thank you for your interesting paper.    Please see the reviews from the reviewers below and take on board their suggestions to further consolidate the paper.     Please put (IDIs) after In depth interviews in abstract.   On page 4 please remove the capital c on confidence interval and check for other capital letter typos.   It would be good to highlight the topics of the qualitative interviews.    I would also like you to engage in more depth with COREQ for qualitative interviews and also discuss quality and rigor in qualitative interviews.    What was the range and mean of interview times for the women.    Please also see further reviews below. 

We look forward to receiving your revised manuscript.

Kind regards,

Julia Morgan

Academic Editor

PLOS ONE

Journal requirements: 1. When submitting your revision, we need you to address these additional requirements.Please ensure that your manuscript meets PLOS ONE's style requirements, including those for file naming. The PLOS ONE style templates can be found at https://journals.plos.org/plosone/s/file?id=wjVg/PLOSOne_formatting_sample_main_body.pdf and https://journals.plos.org/plosone/s/file?id=ba62/PLOSOne_formatting_sample_title_authors_affiliations.pdf.

2. Please provide additional details regarding participant consent. In the ethics statement in the Methods and online submission information, please ensure that you have specified (1) whether consent was informed and (2) what type you obtained (for instance, written or verbal, and if verbal, how it was documented and witnessed). If your study included minors, state whether you obtained consent from parents or guardians. If the need for consent was waived by the ethics committee, please include this information.If you are reporting a retrospective study of medical records or archived samples, please ensure that you have discussed whether all data were fully anonymized before you accessed them and/or whether the IRB or ethics committee waived the requirement for informed consent. If patients provided informed written consent to have data from their medical records used in research, please include this information.

Reviewers' comments:

Reviewer's Responses to Questions

**Comments to the Author**

1. Is the manuscript technically sound, and do the data support the conclusions?

Reviewer #1: Yes

Reviewer #2: Yes

2. Has the statistical analysis been performed appropriately and rigorously? 

Reviewer #1: I Don't Know

Reviewer #2: Yes

3. Have the authors made all data underlying the findings in their manuscript fully available?

Reviewer #1: Yes

Reviewer #2: Yes

4. Is the manuscript presented in an intelligible fashion and written in standard English?

Reviewer #1: Yes

Reviewer #2: Yes

5. Review Comments to the Author

Reviewer #1: This is mixed research on the factors associated with commercial sex work among transgender women in Nepal, with an emphasis on the stigmas they experience in this context. Its theme is current and relevant to the health field. The material requires critical review in the “Methods” section, due to the low level of compliance with the STROBE and COREQ checklists. Furthermore, in the “Disucssion” section, the results relating to the characterization of the participants were not discussed and the discussion about the domains of stigma is restricted to one sentence, without any dialogue with the findings of other studies.

Below is a detailed evaluation of the manuscript, highlighting the points in the text that require revision:

Abstract:

The wording of the objective (to explore […] of Nepal along with sex work stigma and their lived experience) is not the same as that presented in the introduction (to explore […] of Nepal along with their experience and stigma associated with it.). I suggest adopting the wording presented in the introduction.

Methods: Critically review the text considering the recommendations of the STROBE and COREQ checklists. In this sense, I present some notes:

The exclusion criteria adopted were missing.

Considering the technique adopted to capture participants, it remained to be explained how many participants were the seeds and how many referral chains were formed. If there were losses or refusals in this process.

What was the calculation adopted to determine how many participants would be interviewed?

Regarding table 1 (Characteristics of participants involved in the in-depth interview (n =17)), I suggest presenting it in the “Results” section.

In the item “Data collection”, the quantitative stage of the research was missing. Was the instrument for this stage ESES? If the qualitative part of the research had 152 participants and the qualitative part had 17 participants, wasn't the quantitative stage the first to be carried out? The reader is therefore confused by the way the information is described in this section (“The data were collected through face-to-face interviews. For the quantitative survey, the interviews were executed using a set of questionnaires divided into three sections”) , it seems that the article is restricted to presenting qualitative data.

When did the quantitative data collection take place? What about qualitative data?

What procedures were adopted to capture the 17 women who were interviewed?

Where were the interviews carried out?

Who conducted the interviews? Did you have any prior training?

Regarding the data collection instrument: the data collected on the sociodemographic characteristics of the participants was missing; the questions adopted in the second section of the instrument were missing; and the Experiences with Sex Work Stigma Scale (ESWS) failed to be described.

Was the ESWS self-completed by the participants or administered by one of the authors?

What were the study variables?

What was the technique used to analyze the qualitative data? I suggest citing the reference used for this purpose.

Results: I suggest reorganizing the text of this section so that the quantitative results are presented first and, subsequently, presenting the qualitative data.

Discussion: The findings regarding the characterization of the participants (age, relationship status and socialization) were not discussed. And the discussion regarding the domains of stigma is restricted to the phrase “a notable proportion of stigmatization was seen across different domains of the sex work stigma scale”, without any dialogue with the results of other studies.

References: I strongly recommend updating most of the references used, since out of a total of 37 references, 23 are publications that are more than 5 years old.

Reviewer #2: This is an important piece of research examining an under-researched topic. The background to your research in detailed and your methods and analysis are also very thorough. However, I would like to understand why you chose the particular research methods and what the limitations might be. I wonder if a more intersectional analysis would add strength to your findings rather than homogenising the participants.

I would also like to see recommendations for improving the lives of these participants linked to policy.

6. PLOS authors have the option to publish the peer review history of their article (what does this mean?). If published, this will include your full peer review and any attached files.

Reviewer #1: No

Reviewer #2: **Yes: **Sally Bashford-Squires

---

## [Author Response · Author response to Decision Letter 0]

8 Oct 2024

Editor comment

Thank you for your interesting paper. Please see the reviews from the reviewers below and take on board their suggestions to further consolidate the paper.

We would like to thank you for your valuable time and suggestions. We have revised the manuscript as per the provided comments. 

Please put (IDIs) after In-depth interviews in the abstract. On page 4 please remove the capital c on confidence interval and check for other capital letter typos. It would be good to highlight the topics of the qualitative interviews.

Thank you for the suggestion, we have revised it accordingly. 

I would also like you to engage in more depth with COREQ for qualitative interviews and also discuss quality and rigor in qualitative interviews. What was the range and mean of interview times for the women.

Thank you for your comment, we have refilled the COREQ checklist, and have provided the details on the qualifications of the interviewers there as well as briefly described the quality of researchers involved in the qualitative interviews in the manuscript under the ‘Data collection’ and ‘Data management and analysis’ sections.

Thank you for the comment, we would like to apologize for missing some formatting. We have revised the manuscript accordingly. Please let us know if we have missed anything else. 

2. Please provide additional details regarding participant consent. In the ethics statement in the Methods and online submission information, please ensure that you have specified (1) whether consent was informed and (2) what type you obtained (for instance, written or verbal, and if verbal, how it was documented and witnessed). If your study included minors, state whether you obtained consent from parents or guardians. If the need for consent was waived by the ethics committee, please include this information. If you are reporting a retrospective study of medical records or archived samples, please ensure that you have discussed whether all data were fully anonymized before you accessed them and/or whether the IRB or ethics committee waived the requirement for informed consent. If patients provided informed written consent to have data from their medical records used in research, please include this information

Thank you for your comments. In the manuscript we have mentioned these ethical concerns in different portions.

In regards to informed consent, in ‘Data collection’ section we had mentioned that all the interviews were conducted after acquiring written informed consent from the participants. In regards to inclusion of minors, in Sampling procedure section we had mentioned that All the transgender women aged 18 and above, residing inside Kathmandu Valley during the time of data collection were eligible, which directly excludes the minors. This was not a retrospective study or any part of medical records. 

We have included the information on informed consent and consent for recording the audio in the Ethical approval section to make it more clear. 

Thank you, we have only included ethical statements in the ‘Method’ section in the revised manuscript. 

Reviewers comments 

Reviewer #1:

This is mixed research on the factors associated with commercial sex work among transgender women in Nepal, with an emphasis on the stigmas they experience in this context. Its theme is current and relevant to the health field. The material requires critical review in the “Methods” section, due to the low level of compliance with the STROBE and COREQ checklists. Furthermore, in the “Discussion” section, the results relating to the characterization of the participants were not discussed and the discussion about the domains of stigma is restricted to one sentence, without any dialogue with the findings of other studies.

Thank you for your valuable time and suggestions. We have revised the STROBE and COREQ checklist. We have revised the manuscript based on your feedback on each section of the manuscript.

Abstract:

The wording of the objective (to explore […] of Nepal along with sex work stigma and their lived experience) is not the same as that presented in the introduction (to explore […] of Nepal along with their experience and stigma associated with it.). I suggest adopting the wording presented in the introduction.

Thank you for highlighting this inconsistency, we have revised the abstract section. 

Methods: 

Critically review the text considering the recommendations of the STROBE and COREQ checklists. In this sense, I present some notes: The exclusion criteria adopted were missing. Considering the technique adopted to capture participants, it remained to be explained how many participants were the seeds and how many referral chains were formed. If there were losses or refusals in this process.

• Thank you for suggesting this missing and unclear information in our manuscript. We had mentioned that “transgender women aged 18 and above, residing inside Kathmandu Valley during the time of data collection were eligible”, however, we feel that this might not bring a clear picture of inclusion and exclusion criteria so we have added additional information for clarity. 

The study employed a snowball sampling technique, starting with 5 initial seeds who were transgender women engaged in social or activist networks. These seeds were asked to refer other transgender women they knew who fit the eligibility criteria. Through this process, a total of 4 referral chains were formed. The Blue Diamond Society supported the data collection and the snowball sampling process. We have included this in the revised manuscript.

What was the calculation adopted to determine how many participants would be interviewed? Regarding table 1 (Characteristics of participants involved in the in-depth interview (n =17)), I suggest presenting it in the “Results” section. 

Thank you for your comment. The number of qualitative interview participants was determined based on the principle of data saturation. We have moved the ‘Characteristics of participants involved in the in-depth interview (n =17)’ into the result section as suggested.

In the item “Data collection”, the quantitative stage of the research was missing. Was the instrument for this stage ESES? If the qualitative part of the research had 152 participants and the qualitative part had 17 participants, wasn't the quantitative stage the first to be carried out? The reader is therefore confused by the way the information is described in this section (“The data were collected through face-to-face interviews. For the quantitative survey, the interviews were executed using a set of questionnaires divided into three sections”) , it seems that the article is restricted to presenting qualitative data.

Thank you for helping us bring clarity in our manuscript. We have revised the ‘Data Collection’ section to clarify the following: “Both the quantitative and qualitative phases of data collection were conducted through face-to-face interviews. For the quantitative phase, a structured questionnaire with close-ended questions was used, while for the qualitative phase, an open-ended interview guide was employed. This study followed a concurrent mixed-methods design, meaning that both the quantitative and qualitative data were collected simultaneously. Participants for the qualitative interviews were recruited during the quantitative phase. Experiences with Sex Work Stigma Scale (ESWS) was used in the quantitative phase.” We hope this clarifies the structure of the study and addresses the confusion.

When did the quantitative data collection take place? What about qualitative data? What procedures were adopted to capture the 17 women who were interviewed? Where were the interviews carried out? Who conducted the interviews? Did you have any prior training?

Thank you for suggesting the missing information in our manuscript, we have revised the section to make both phases clear and added the missing information that has been suggested. All interviews were conducted in private settings at the residences of the TGWs by HSP, a public health professional with prior training. 

Regarding the data collection instrument: the data collected on the sociodemographic characteristics of the participants was missing; the questions adopted in the second section of the instrument were missing; and the Experiences with Sex Work Stigma Scale (ESWS) failed to be described. Was the ESWS self-completed by the participants or administered by one of the authors? What were the study variables? What was the technique used to analyze the qualitative data? I suggest citing the reference used for this purpose.

Thank you for your comment. All the data, including ESWS, were collected through face-to-face interviews. We have revised the manuscript to make it clear. We are unclear on your comment regarding “the data collected on the sociodemographic characteristics of the participants was missing; the questions adopted in the second section of the instrument were missing; and the Experiences with Sex Work Stigma Scale (ESWS) failed to be described”, as we have mentioned the various sections of the quantitative tool in the ‘Data collection’ section and the collected variables are presented in the first table of the result section. We have also mentioned what the second section of the data collection tool covered. We have described ESWS along with its citation. We are more than happy to attach the questionnaire used in the study as a supplemental file if needed. The technique and process used in qualitative data analysis has been described under the ‘Data management and analysis’ section.

Results: I suggest reorganizing the text of this section so that the quantitative results are presented first and, subsequently, presenting the qualitative data. 

Thank you for your suggestion, we have revised the manuscript accordingly. 

Discussion: The findings regarding the characterization of the participants (age, relationship status and socialization) were not discussed. And the discussion regarding the domains of stigma is restricted to the phrase “a notable proportion of stigmatization was seen across different domains of the sex work stigma scale”, without any dialogue with the results of other studies.

The characterization of the participants has been explained in the results section of our study. We are a bit unclear regarding your comment concerning not including discussion about characterization of the participants (age, relationship status and socialization). We would be happy to revise our discussion as per your suggestion if you could guide us on what you meant. Currently our discussion is more focused on the factors found to be associated with CSW, based on our study objectives. 

As there are limited studies that have assessed Commercial sex work stigma and during the literature review, we were unable to find any studies that have focused on Transgender women and their stigma related to CSW. Most of the literatures on CSW stigma are connected with HIV, which is not a focus of our study. So, we feel that it will not be relevant and not compatible to make a discussion on CSW stigma based on Female or male gender and make its reflection to TWGs considering their unique challenges and context. Furthermore, in our result section we have explored the CSW stigma among TWGs in detail so we felt that repeating the findings from the results in discussion might be redundant. 

References: I strongly recommend updating most of the references used, since out of a total of 37 references, 23 are publications that are more than 5 years old.

Thank you for your suggestion. We acknowledge that there are studies that are older than 5 years. Some of the older studies are references to legal documents and most of the studies have focused on transgender women, sex work, and sex work among transgender women which do not have many recent studies and the literatures used in our manuscript cannot be replaced as the reflections are drawn from these literatures. As there are limited studies on this topic we have included literature that we found relevant to our study. We feel that the original studies should be acknowledged.

Reviewer #2: 

This is an important piece of research examining an under-researched topic. The background to your research in detailed and your methods and analysis are also very thorough. However, I would like to understand why you chose the particular research methods and what the limitations might be. I wonder if a more intersectional analysis would add strength to your findings rather than homogenising the participants. I would also like to see recommendations for improving the lives of these participants linked to policy.

We chose a quantitative component to understand the magnitude of the problem (i.e. commercial sex work among TWG) and a qualitative component to explore their lived experience including their sex work stigma. Thank you for your comment regarding intersectional analysis, however, considering the small sample size we think we are unable to go for intersectional analysis at this stage. We acknowledge the value of this approach and have included it as a suggestion for future research, where a larger and more diverse sample could enable a robust intersectional analysis.

Thank you for notifying this missing component, we have added some recommendations for improving the lives of these participants linked to policy in our discussion section.

---

## [Decision Letter · Decision Letter 1]

8 Nov 2024

PONE-D-24-19414R1Exploring Commercial Sex Work among Transgender Women in Nepal: Contributors and Stigma - A Mixed-Method StudyPLOS ONE

Dear Dr. Paudel,

Thank you for submitting your manuscript to PLOS ONE. After careful consideration, we feel that it has merit but does not fully meet PLOS ONE’s publication criteria as it currently stands. Therefore, we invite you to submit a revised version of the manuscript that addresses the points raised during the review process.

Please consider the reviewers comments below and make changes.   In addition please consider the following: 1) Please be consistent with the use of TGW and CSW throughout sometimes it is written as Transgender women and other times TGW, the same with commercial sex work2) Please use the term commercial sex work rather than prostitution3) In the Introduction take out reference to trans-sexual - use transgender only4) In the introduction you write 'in comparison to female sex workers' maybe consider using the term 'cis-gendered women' rather than female. 

We look forward to receiving your revised manuscript.

Kind regards,

Julia Morgan

Academic Editor

PLOS ONE

Journal Requirements:

Reviewers' comments:

Reviewer's Responses to Questions

**Comments to the Author**

1. If the authors have adequately addressed your comments raised in a previous round of review and you feel that this manuscript is now acceptable for publication, you may indicate that here to bypass the “Comments to the Author” section, enter your conflict of interest statement in the “Confidential to Editor” section, and submit your "Accept" recommendation.

Reviewer #1: (No Response)

2. Is the manuscript technically sound, and do the data support the conclusions?

Reviewer #1: Yes

3. Has the statistical analysis been performed appropriately and rigorously? 

Reviewer #1: Yes

4. Have the authors made all data underlying the findings in their manuscript fully available?

Reviewer #1: Yes

5. Is the manuscript presented in an intelligible fashion and written in standard English?

Reviewer #1: Yes

6. Review Comments to the Author

Reviewer #1: The authors met the recommendations and suggestions from the last round of evaluation, except:

Although the text presents the steps of the analytical process of qualitative data, the authors did not clarify which analysis technique was used or include a reference.

References published more than 5 years ago predominate.

7. PLOS authors have the option to publish the peer review history of their article (what does this mean?). If published, this will include your full peer review and any attached files.

Reviewer #1: No

---

## [Author Response · Author response to Decision Letter 1]

10 Nov 2024

Dear Editor, 

We sincerely thank you and the reviewers for your valuable feedback and time dedicated to improving our manuscript. We have carefully considered each comment and have made revisions to address these suggestions. Below is a detailed response to each point raised:

Editor comment and author response 

1) Please be consistent with the use of TGW and CSW throughout sometimes it is written as Transgender women and other times TGW, the same with commercial sex work

We have ensured consistent use of terminology throughout the manuscript. Specifically, we have spelled out “Transgender women” and “commercial sex work/commercial sex workers” throughout the document to enhance clarity for readers and avoid any potential confusion. We have refrained from using the abbreviations TGW and CSW.

2) Please use the term commercial sex work rather than prostitution

We have replaced the term "prostitution" with "commercial sex work" throughout the manuscript.

3) In the Introduction take out reference to trans-sexual - use transgender only

In the introduction, we have removed the term "trans-sexual" and used "transgender" exclusively, as suggested.

4) In the introduction you write 'in comparison to female sex workers' maybe consider using the term 'cis-gendered women' rather than female. 

Thank you, We have replaced “female sex workers” with “cis-gendered women” to align with more accurate and inclusive terminology.

Reviewers comment and author response 

Although the text presents the steps of the analytical process of qualitative data, the authors did not clarify which analysis technique was used or include a reference.

We appreciate the time and effort that has been dedicated to strengthening our work. Thank you for notifying the missing information, we have now clarified that thematic analysis was used, along with an appropriate citation detailing the specific steps followed.

References published more than 5 years ago predominate.

We acknowledge the feedback regarding the use of references predominantly older than five years. Due to the specificity of our research topic, recent studies are limited, and many current publications refer back to foundational sources. To ensure accuracy, we chose to cite original sources where relevant rather than secondary citations in more recent publications. We hope this approach is understandable given the context of our study. Thank you once again for the opportunity to improve our manuscript.

---

## [Editor Report · Decision Letter 2]

14 Nov 2024

Exploring Commercial Sex Work among Transgender Women in Nepal: Contributors and Stigma - A Mixed-Method Study

PONE-D-24-19414R2

Dear Dr. Paudel,

We’re pleased to inform you that your manuscript has been judged scientifically suitable for publication and will be formally accepted for publication once it meets all outstanding technical requirements.

Kind regards,

Julia Morgan

Academic Editor

PLOS ONE
---

## [Editor Report · Acceptance letter]

18 Nov 2024

PONE-D-24-19414R2 

PLOS ONE

Dear Dr. Paudel, 

I'm pleased to inform you that your manuscript has been deemed suitable for publication in PLOS ONE. Congratulations! Your manuscript is now being handed over to our production team.

Kind regards, 

on behalf of

Dr. Julia Morgan 

Academic Editor

PLOS ONE